# Associations between Patient Report of Pain and Intervertebral Foramina Changes Visible on Axial-Loaded Lumbar Magnetic Resonance Imaging

**DOI:** 10.3390/diagnostics12030563

**Published:** 2022-02-23

**Authors:** Tomasz Lorenc, Marek Gołębiowski, Dariusz Syganiec, Wojciech M. Glinkowski

**Affiliations:** 1Ist Department of Clinical Radiology, Medical University of Warsaw, 02-091 Warsaw, Poland; tlorenc@wum.edu.pl (T.L.); marek.golebiowski@wum.edu.pl (M.G.); dariusz.syganiec@uckwum.pl (D.S.); 2Department of Medical Informatics and Telemedicine, Center of Excellence “TeleOrto” for Telediagnostics and Treatment of Disorders and Injuries of the Locomotor System, Medical University of Warsaw, 00-581 Warsaw, Poland

**Keywords:** lumbar intervertebral foramen, foramen area, stenosis, magnetic resonance imaging, axial loading, in vivo, lumbar spine, dermatome, low back pain

## Abstract

The intervertebral foramen may influence spinal nerve roots and, therefore, be related to the corresponding dermatomal pain. In vivo evaluation of the intervertebral foramen–dermatome relationship is essential for understanding low back pain (LBP) pathophysiology. The study aimed to correlate the lumbar MRI unloaded-loaded foraminal area changes with dermatomal pain in the patient’s pain drawings. Dynamic changes of the dermatomal pain distribution related to the intervertebral foramen area changes between quantitative conventional supine MRI (unloaded MRI) and axial-loading MRI (alMRI) were analyzed. The MRI axial-loading intervertebral foramen area changes were observed, and the most significant effect of reducing the foraminal area (−6.9%) was reported at levels of L2–L3. The incidence of pain in the dermatomes increases linearly with the spine level, from 15.6% at L1 to 63.3% at L5 on the right and from 18.9% at L1 to 76.7% at L5 on the left. No statistically significant effect of changes in the intervertebral foramen area on the odds of pain along the respective dermatomes was confirmed. Changes in the foraminal area were observed between the unloaded and loaded phases, but differences in area changes between foramen assigned to painful dermatomes and foramen assigned to non-painful dermatomes were not significant.

## 1. Introduction

The differential diagnosis of low back pain includes a wide range of diagnoses that may result in irritation of the spinal nerve roots in the lower lumbar spine due to the pathology of the intervertebral foramen [1,2,3,4,5,6]. Spinal nerves run from the spinal cord to the peripheral effectors in the motor range and from the skin centripetally through the intervertebral foramen. The course of the spinal nerves and the corresponding skin sensation areas are well defined, and their surface location has been systematically examined [7,8]. Dermatomes are areas of the skin of the trunk and extremities innervated by the cutaneous branches of the dorsal and ventral branches of the spinal nerves [8,9,10]. Each of these nerves transmits sensations, including pain, from a specific skin area to the brain, where the right spinal root L1 passes through the right intervertebral foramen L1–L2 and delivers the right dermatome L1. The scheme repeats at levels lower down to L5–S1 [7]. Therefore, the pathology of the L1–L2 intervertebral foramen may irritate the L1 spinal nerve and thus may be reflected in pain within the L1 dermatome, which also applies to other spinal levels.

The intervertebral foramen is crucial in low back pain development studies [11,12]. In recent years, with the rapid development of minimally invasive spine surgery, percutaneous foraminal endoscopy has provided a less invasive technique to address neuroforaminal pathology [13]. Nevertheless, surgical procedures in the vicinity of the spinal root and the dorsal root ganglia should be minimized due to their sensitivity to mechanical pressure and the risk of possible worsening or development of postoperative symptoms, such as pain. In the scope of recent surgical advances, especially minimally invasive techniques and spine endoscopy, it is challenging to ignore the importance of foraminal pathology. Magnetic resonance imaging (MRI) proves to be an excellent method for assessing the condition of the intervertebral foramen, determining the degree of compression of the nerve root and the anatomical cause in the spine, especially in the lumbosacral section. Diagnostic imaging of the intervertebral foramen pathology is performed using multiple MRI sequences, including a simple standard examination and new protocols, such as diffusion tensor imaging, ultra-short echo time, and T2 mapping [14,15]. However, the relations between anatomical impairment and LBP symptoms, with few exceptions, remain speculative and controversial; e.g., critical narrowing of the neural foramina, inapparent in conventional magnetic resonance (MR) studies, may occur when loads are applied to the spine, and clinically speculated results do not necessarily match the MRI findings for nerve root compression [16,17]. However, it should be noted that a conventional MRI system cannot fully achieve this goal, as it can only examine patients in a supine position, while an examination with compressive forces requires an upright position. Axial-loading MRI is a feasible diagnostic tool for simulating the spine under physiological conditions [18,19,20]. Some studies suggest that altered biomechanics at the IF level are responsible for developing instability of adjacent segments, degeneration, and stenosis [21,22]. Intervertebral instability is expected to be associated with a higher incidence of low back pain than normal and reflected in dynamic changes visualized by axially loaded MRI [18,19].

On the other hand, normal intervertebral foramen presents physiologic soft tissue compression upon loading, which results from average elasticity; therefore, increased stiffness of the foraminal zone is rather observed as the last stage of self-limiting instability as well as disc dehydration or progressive degenerative changes, causing stabilization of the motion segments [20,23]. In light of recent updates, we decided to explore the in vivo biomechanics of the intervertebral foramen, focusing on dermatomal pain development. Furthermore, despite progress in evaluating the foraminal zone and the development of low back pain, the literature has not reported on any morphologically derived quantitative metrics of the intervertebral foramen concerning a dermatomal pattern that could be used to aid in the classifications of spine-related pain. The degenerated and non-degenerated changes in the lumbar intervertebral foramina under physiological loading conditions and the relationship with the distribution of pain along the dermatomes are unknown. The experimental work presented here provides one of the first investigations into how changes in the foraminal area affect dermatomal pain. The authors hypothesized that altered biomechanics at the foramen level could be responsible for the development of pain along the dermatome supplied by a related nerve root that exits through this altered foramen. This study tests the hypothesis of whether dermatomal pain along a particular nerve root correlates with the area change of the respective intervertebral foramen observed between recumbent and axially loaded MRI.

## 2. Materials and Methods

The study was an observational study conducted on 90 consecutive patients referred for a lumbar spine MRI with lower back pain as an indication. Due to the research protocol, patients were consecutively examined following the National Health Fund waiting list for diagnostic imaging examinations and following the diagnostic workflow with no priorities. Exclusion criteria included significant spinal deformity or fracture, osteoporosis, previous spine surgery, lack of patient compliance, body mass less than 40 kg, and lack of written consent from the patient. General contraindications to MRI examinations (e.g., pacemakers, ferromagnetic implants, foreign bodies, and claustrophobia) were also considered.

### 2.1. Axially Loaded MRI

The examination was performed using a 1.5 T MRI (Ingenia, Philips Healthcare, Eindhoven, The Netherlands). Axial-loading MRI was applied using a compression device with an external nonmagnetic DynaWell (DynaWell L-Spine, DynaWell Diagnostics, Las Vegas, NV, USA). The 3D T2-weighted volume isotropic turbo spin echo acquisition (VISTA) utilized for the present study were acquired with the following parameters: average repetition time 2000 ms, average echo time 90 ms, number of signals averaging 1, acquisition voxel 1.0 × 1.0 × 0.5 mm, reconstruction matrix 640, reconstruction voxel 0.47 × 0.47 × 0.5 mm, turbo factor 61, and average scan time 6 min. First, images in the recumbent position were acquired; then, the VISTA sequence was repeated under axial load. According to previous disc pressure measurements [24], the chosen load was equal to 40–50% of the patients’ body weight, with the same load distribution in both legs (20–25% of body mass per leg). The patient was subjected to this load in the lying position for at least 5 min before examination (Figure 1).

### 2.2. Image Analysis

The images were evaluated in a single center on a dedicated workstation (IntelliSpace Portal, Philips Healthcare, Eindhoven, The Netherlands). The sagittal cross-section area of the vertebral foramina was determined for each level on both sides, from L1–L2 to L5–S1. Measurements were made by encircling the vertebral foramina area in sagittal cross-sections at the same levels for the phase with and without axial loading (Figure 2). The foraminal area was defined as the area bounded by the adjacent superior and inferior vertebral pedicles, the posterosuperior boundary of the inferior vertebral body, the surface of the intervertebral disc anteriorly, the posteroinferior boundary of the superior vertebral body, and the surface of the ligamentum flavum posteriorly. All foraminal area (FA) measurements were made by two independent reviewers (T.L. and D.S.) who were blinded to clinical outcomes. The FA values used in the statistical analysis represent the average of the values calculated by the two reviewers. In cases where the scores differed by more than 10%, the value used for analysis was the consensus of two reviewers’ measurements.

### 2.3. Self-Report Measure of Pain Distribution

On arrival for their scheduled MRI, participants completed a document asking them to fill in a body drawing depicting the typical anatomic location of their pain. Pain location was determined using patient pain drawings: an outline of a human figure on which the patient marks the areas where they experience pain [25]. Patients with nonmalignant pain had shaded in their experienced pain in the front and back views of a pain drawing. The dermatomal system was used to classify patients by pain distribution from body drawing. The transparent grids were placed over the pain drawing for analyzing the pain dermatomes and classifying the scores. We scored the pain areas in the low back and legs according to the different nerve roots from L1 to L5, bilateral. The completed drawings were scored for the presence or absence of pain in the dermatomes.

### 2.4. Statistical Methods

The dependence of the incidence of pain along the dermatomes on the magnitude of the load-induced changes in the intervertebral foraminal space was analyzed. The areas of each lumbar intervertebral foramen (L1–L2, L2–L3, L3–L4, L4–L5, and L5–S1) were measured independently before and after loading by two investigators. The mean values of these measurements were used to calculate the percentage changes of the intervertebral foramen area. The generalized estimating equation (GEE) method and the generalized linear model (GLM) were used to analyze the dependence of pain occurrence on changes in the size of each intervertebral foramen. A binary matrix variable, “Dermatomal pain” (0, No Pain; 1, Pain), was defined as the dependent variable describing the presence of pain in the dermatome area associated with the corresponding intervertebral foramen. As independent variables, the model included a matrix variable, “Mean percentage changes of intervertebral foramen area”, that specifies the percentage changes in the intervertebral foramen area for each level and side of the spine. A negative value of changes in the intervertebral foramen area (%) means reduced foramen, while a positive value means that the foramen is enlarged with axial loading. The variables were categorized into specific equinumerous levels to allow a nonlinear relationship between intervertebral foramen area changes and the odds of pain occurrence. Three nearly equinumerous levels, namely <−7% (foramen with a diminished area of more than seven percent); −7–0% (foramen with a diminished area of less than seven percent); and ≥0% (foramen with an enlarged area after axial loading), were set. All tests were performed at the level of statistical significance of 0.05. The IBM SPSS Statistics 23.0 package was used for statistical analysis.

## 3. Results

The estimated incidence of pain in the dermatome on both sides increased linearly with the level of the spine. Pain incidence increased from 15.6% for L1 to 63.3% for L5 on the right and 18.9% for L1 to 76.7% for L5 on the left side. Table 1 shows the distribution of dermatome pain by side and level of the spine. Figure 3 represents cumulative pain drawings that demonstrate the location of pain in patients within the distribution of the L1–L5 dermatome. 

Figure 4 shows the average percentage changes in the intervertebral foramen area by the side and the spine level. Furthermore, in this case, the pictures for the right and left sides are similar, but the dependence on the spine level is more square than linear. A negative change value means a reduction in the foramen area after loading and a positive increase in the foramen area. The most noticeable reduction of the foraminal area for both sides was found at the level L2–L3 and was −6.9% for both the right and left sides. Table 2 presents the mean changes in the intervertebral foramen area for all levels and on both sides.

As shown in Table 3, the study group was almost equinumerous in subgroups of changes in the intervertebral foramen area.

There was no statistically significant effect of changes in the intervertebral foramen area on the odds of pain incidence in the area of the dermatomes. A statistically significant influence (*p* < 0.001) of the level and side of the spine on the odds of pain was found. The odds of pain increase with the spine level, roughly doubling with each level, and are 1.6 times higher on the left side than on the right side. Table 4 presents the parameters of the GLM model.

## 4. Discussion

This study reports the relationship between dermatomal pain and changes in the foraminal area diagnosed by axial-loading magnetic resonance imaging of the lumbar spine. To our knowledge, this is the first study to use axial loading to measure the difference between loaded-unloaded in the foraminal area to correlate it with pain along with the spinal roots that exit the related neuroforamina. Contrary to expectations, this study did not find a significant dependence of changes in the intervertebral foramen area on pain occurrence in the respective dermatomes. In this study, no correlations were found between the size of the changes in the foraminal area between recumbent and axial loading and the appearance of dermatomal pain along with the accompanying nerve roots. Although the general pattern of dermatomes is similar in all people, the precise areas of innervation are unique to an individual. Charts of dermatomal sensory distributions have been made from experimental studies. However, there is considerable variability in their results [26]. Muscles can also have distinctive pain patterns (myotomes), as can skeletal structures (sclerotomes), due to these structures arising from different embryonic tissues, which patients could also superimpose on dermatomal pain patterns [10,27]. Future neurophysiological assessment should correlate foraminal dynamics and pain with spinal roots [8].

Nevertheless, pain drawing has become a widely used method for gaining patients’ perspectives in the subjective assessment of pain [25]. A growing interest is observed in drawing charts applications for marking pain areas and other sensory impairments [28]. This study has shown that the estimated incidence of pain in dermatomes increases linearly with the spine level, from 15.6% at L1 to 63.3% at L5 on the right and from 18.9% at L1 to 76.7% at L5 on the left. The chance of pain increases with the spine level, roughly doubling with each level (Figure 3). The increase in the chance of pain location more and more caudally can be explained by significantly often-observed changes in the intervertebral discs and intervertebral foramen at L4–L5 and L5–S1 [29]. Travis Caton Jr. et al. [29] found that the prevalence of significant neuroforaminal stenosis (NFS) and spinal canal stenosis (SCS) increased caudally from T12–L1 to L4–L5, and younger patients (<50 years) had a relatively higher prevalence of NFS at L5–S1.

Interestingly, in our study, the estimated incidence of pain in dermatomes is 1.6 times higher on the left side than on the right side. The reasons for the presentation of asymmetric pain remain unclear. The location of the abdominal aorta on the left side and the disturbances in blood flow in its lumen are considered potential mechanisms of the occurrence of asymmetric changes in the intervertebral discs and, consequently, in the intervertebral foramen [30]. The intensification of changes in the pressure wave in the aorta may be influenced by age, and the presence of atherosclerotic plaques or aortic bifurcation topography may affect the pressure wave transmitted through the aorta, which could secondarily predispose to asymmetric degeneration [31,32,33]. Another potential cause is the relatively common occurrence of lumbosacral transitional vertebrae (even over 30% of the population) [34,35,36]. Another considered mechanism of pain asymmetry may be a coronal imbalance in the course of scoliosis [37]. Some authors mention the dominant side handedness in the population that may lead to the imbalanced use of paraspinal muscles and the axial skeleton observed in scoliosis and imbalanced occupation, recreational, or sports activities [38,39]. The above-described mechanisms may also explain the symptoms asymmetry seen in our study.

This study displayed dynamical properties of the intervertebral foramen by quantitative MRI of the intervertebral foramen performed with conventional supine MRI (unloaded MRI) and axial loading during MRI (alMRI). The most significant effect of reducing the foraminal area was observed at the level of L2–L3, and it was −6.9% for both the right and the left sides (Figure 5). The intervertebral foramen area increased by 2.25–2.48% at the level L5–S1 when exposed to axial load (Figure 6). Besides, the current study results indicate that researching the future with potentially better resolution and specificity on the image-based characterization of the intervertebral foramina could further support diagnostic quality and surgical decision making. The study showed that a high-resolution 3D MRI is feasible under axial compression. Volume ISotropic Turbo spin-echo Acquisition techniques have been used to acquire high-resolution, contiguous, thin-section isotropic images for complex spine anatomy. In our opinion, high-resolution 3D imaging will make automatic image recognition more accurate. A computational approach in automatic image recognition based on machine learning and deep learning will ease radiological measurements of the lumbar spine. Artificial intelligence in image recognition and segmentation is desirable to automate the evaluation of the lumbar spine and obtain a good level of clinical prediction.

The study addresses the correlation of clinical symptoms with magnetic resonance imaging. Therefore, its use in the triage of appropriate clinical referrals is limited. The outlined findings motivate further research of the dynamics of the intervertebral foramen concerning the root ganglia and other critical neuroforaminal structures.

The position of the ganglion changes from within the intervertebral foramen to the spinal canal as one moves to the L5 nerve root [40]. Furthermore, future work can use these findings to assess the relationship of other intervertebral foramen structures with symptoms and the value of alMRI in a group of asymptomatic individuals.

Compared with previous studies on the association between dermatomal pain and axial-loading foraminal changes, this is somewhat complicated due to the availability of only porcine models or weight-bearing MRI studies. Previous porcine model studies evaluated the percent occlusion of the intervertebral foramen with different load protocols applied to human lumbar spine specimens. Cuchanski et al. [21] observed that the mean values of intervertebral foramen occlusion under a 250 N axial load were 7.8% ± 4.7%. These results seem to be consistent with our research. Fujiwara et al. [22] studied mechanisms involved in changing foraminal dimensions through axial rotation of the lumbar spine specimens. These movements resulted in a 5.7% decrease in the cross-sectional foraminal area.

The dimensions and anatomical relationships of the neural foramina are constantly changing during normal daily activities. Therefore, foraminal pathology may be an intermittent and dynamic process in some spines. Previous studies noted as weight-bearing did not focus on spinal load but assessed the importance of flexion and extension of the spine and its influence on intervertebral foramina. Singh et al. [12] showed that the average percent decrease in the foraminal area was 30.0%, with the most significant decrease from flexion to extension occurring at L2–L3 (35.7%) and the minor change occurring at L5–S1(21.5%). It differs from the findings presented here, where we observed the most significant effect of reducing the foraminal area for both sides at the L2–L3 level, and it was 6.9%; and, in contrast to previous findings, the area of intervertebral foramen increased by 2.25–2.48% at the level of L5–S1 when exposed to axial loading. Differences between flexion-extension kinematic MRI and axial-loading MRI may have influenced the range of area changes with the more significant variation reported in flexion-extension kinematic MRI. Besides, similarities are observed between both modalities, with the most significant area decrease occurring at L2–L3. The study conducted by Splendiani et al. [11] has evaluated the presence of dynamic foraminal stenosis using a low-field magnetic resonance unit that allows images to be acquired both in the recumbent and upright position. MR examinations under weight-bearing conditions (orthostatic position) could detect the presence of dynamic stenosis of the intervertebral foramen in 61/230 levels compared to zero cases in a recumbent position. Morphological characterization of the state of the intervertebral foramina under examination was performed based on a visual scale and classified as stenotic, dubious, or nonstenotic. Unfortunately, the authors did not perform measurements of the intervertebral foramina. It should be considered that the images in orthostatic conditions, performed utilizing the low-field machine (0.25 T), have a less favorable signal-to-noise ratio than those obtained in clinostatic conditions, obtained employing high-field MR with axial load.

More recently, minimally invasive percutaneous endoscopic lumbar foraminoplasty has been proposed, which could use the anatomical and functional details given by both pretreatment and posttreatment testing to obtain a more in-depth understanding of the phenomena underlying such a widespread pathology with such high social costs [13]. Using tubular retractors with endoscopy achieves focal decompression, especially when treating foraminal pathology in older patients [13]. Yeung and Gore stated that foraminal stenosis would be one of the main pathologies for failed back syndrome [41]. However, percutaneous endoscopic lumbar foraminoplasty is bound to cause a risk of an iatrogenic nerve root or dural sac injury, and excessive resection of the superior articular process can cause potential postoperative low back pain and lumbar segmental instability [42]. The prompt diagnosis of the site and characteristics of the foraminal pathology may provide helpful guidance for the correct treatment procedures.

There were several limitations to the present study. The retrospective approach may provide a possibility of bias. The enrollment in this study consisted of patients with LBP and LBLP symptoms. Another limitation may derive from the time gap between the referral physician’s examination and the date of MRI imaging, owing to no strict neurological examinations being performed at the time of MRI. Another potential source of weakness in this study was the lack of a computational approach in automatic image recognition based on machine learning and deep learning to facilitate radiological measurements of the lumbar spine.

## 5. Conclusions

In conclusion, the appearance of back pain along the L1–L5 dermatomes is not simply or directly related to the percentage difference in the area of the intervertebral foramen observed between loading and unloading. Nevertheless, further research should be conducted to explain dermatomal pain beyond the increased stiffness of the foraminal zone or simple nerve compression by a hyperdynamic foramen. It was observed in the study that MRI with axial loading makes possible the measurements and detection of changes in the intervertebral foramen anatomy. These findings suggest that further research is warranted to determine the potential utility of axial-loading MRI in clinical decision making.

## Figures and Tables

**Figure 1 diagnostics-12-00563-f001:**
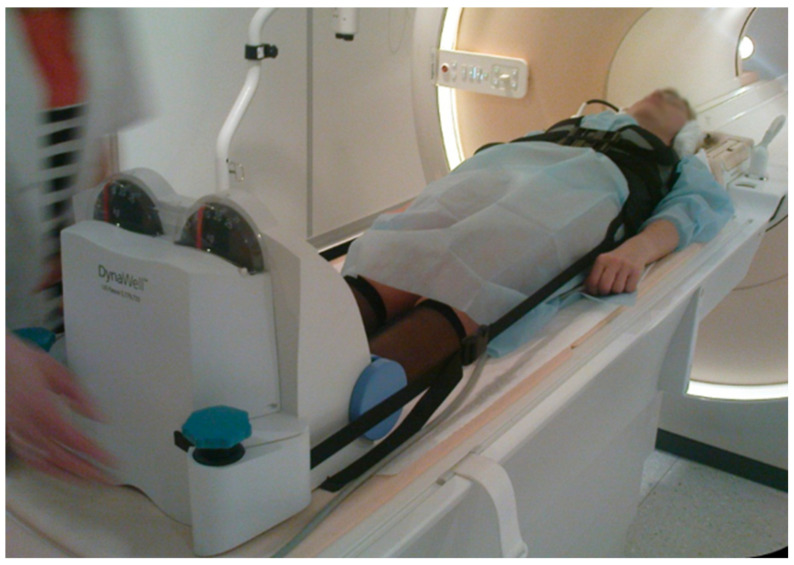
A patient on the MRI scanner with the compression device (DynaWell). A harness is attached with straps to a footplate, applying an axially directed load.

**Figure 2 diagnostics-12-00563-f002:**
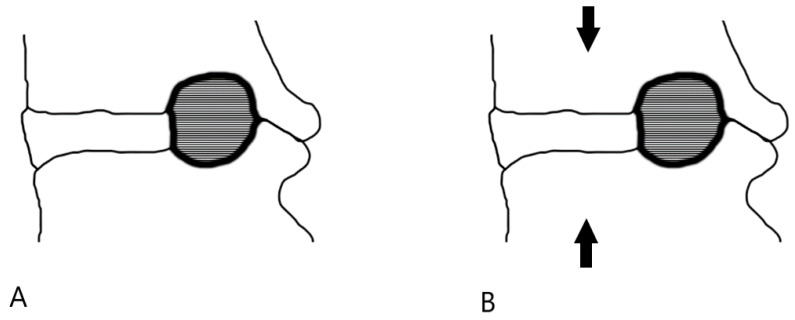
Image analysis. The sagittal cross-section area of the vertebral foramen without axial loading (**A**) and after axial loading (**B**).

**Figure 3 diagnostics-12-00563-f003:**
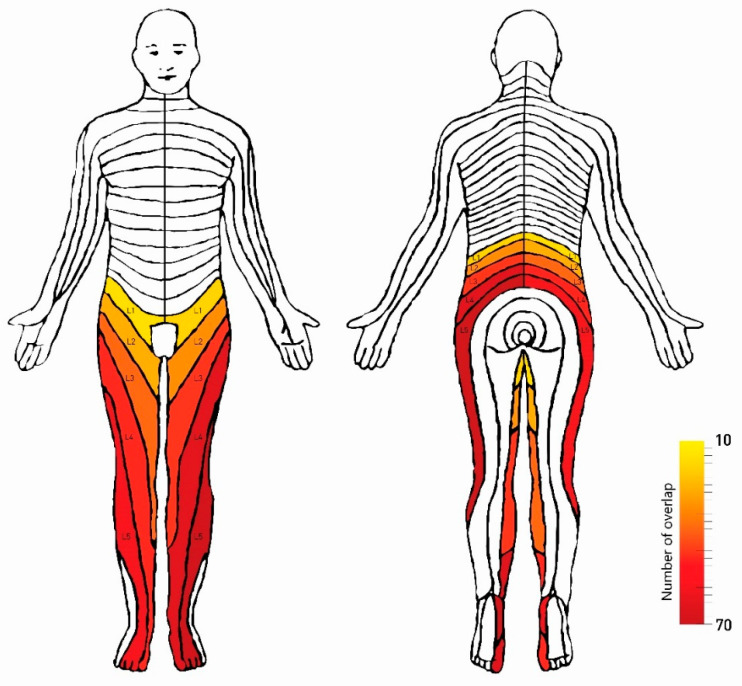
Cumulative pain distribution based on patient perception of pain within L1–L5 dermatomes in a final group of 90 patients.

**Figure 4 diagnostics-12-00563-f004:**
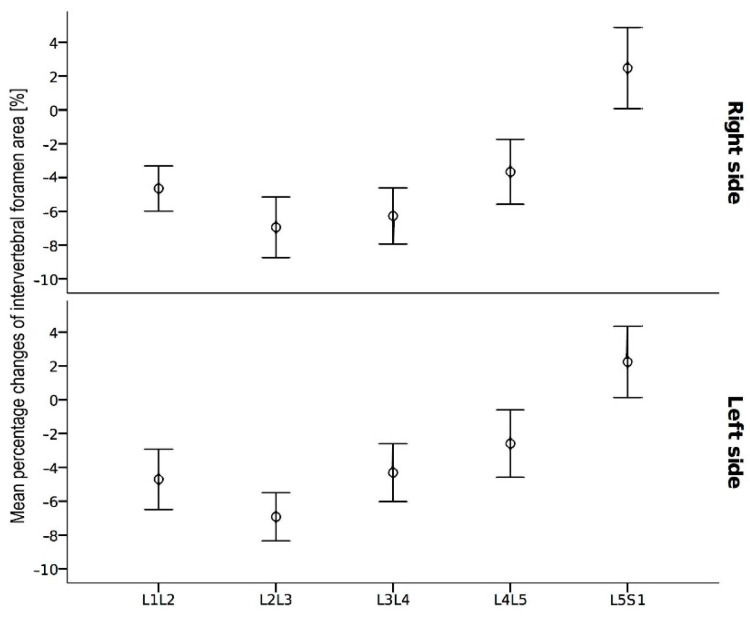
Error bar plot of the loaded-unloaded changes in the intervertebral foramen area at each vertebral level on the right side (**top**) and the left side (**bottom**). Circles represent mean values of the percentage changes in the area of the intervertebral foramen, and the vertical line segments indicate a 95% confidence interval.

**Figure 5 diagnostics-12-00563-f005:**
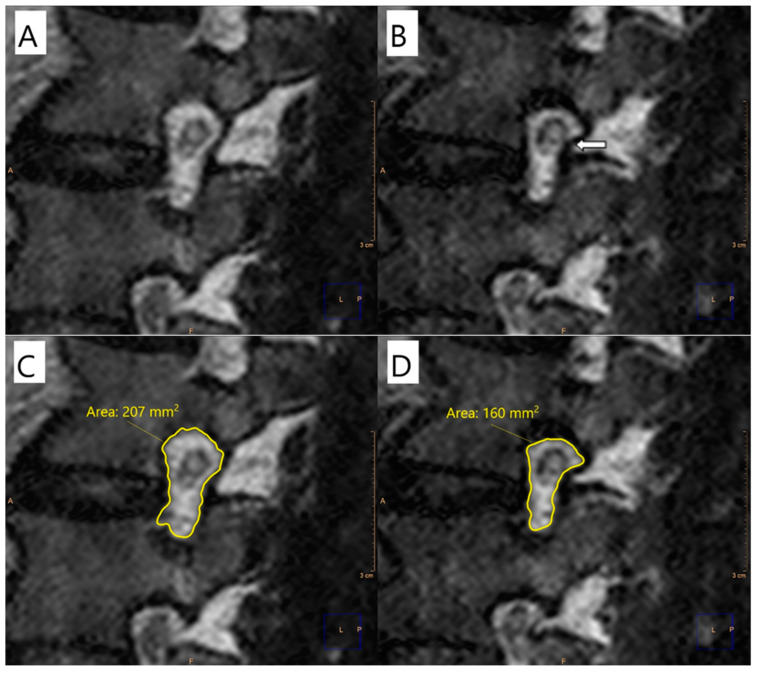
Sagittal magnetic resonance cross-section through the L2–L3 foramen. (**A**) Unloaded MRI shows normal relationships between the foramen and surrounding structures with no perineural fat obliteration nor root morphological changes; (**B**) axial-loaded MRI shows perineural fat obliteration surrounding the nerve root in the transverse direction (arrow); changes in the area of the intervertebral foramen at L2–L3 seen without (**C**) and with (**D**) axial loading.

**Figure 6 diagnostics-12-00563-f006:**
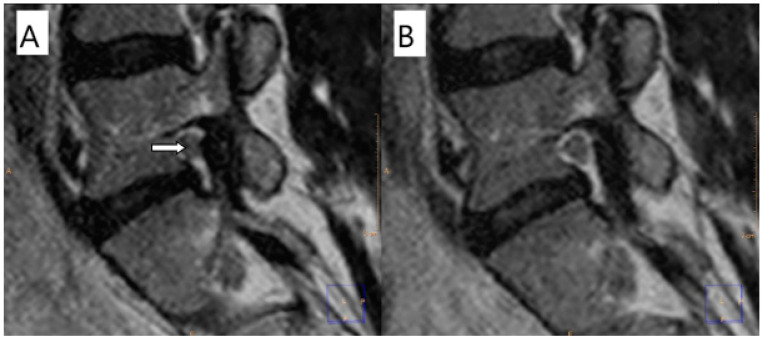
Sagittal magnetic resonance cross-section through the L5–S1 foramen. (**A**) Unloaded MRI shows perineural fat obliteration surrounding the nerve root in the transverse direction and root morphologic change (arrow). (**B**) Axial-loaded MRI shows normal relationships between foramen and surrounding structures with no perineural fat obliteration nor root morphologic changes. (**C**) The image shows the intervertebral foramen area at L5–S1 without axial loading. (**D**) The image presents the L5–S1 intervertebral foramen area with axial loading.

**Table 1 diagnostics-12-00563-t001:** Distribution of dermatomes by pain, side, and level.

Dermatomes (Side/Level)	Right		Left	
	Pain	No Pain	Pain	No Pain
L1	14 15.6%	76 84.4%	17 18.9%	73 81.1%
L2	25 27.8%	65 72.2%	31 34.4%	59 65.6%
L3	38 42.2%	52 57.8%	46 51.1%	44 48.9%
L4	48 53.3%	42 46.7%	63 70.0%	27 30.0%
L5	57 63.3%	33 36.7%	69 76.7%	21 23.3%
Total	182 40.4%	268 59.6%	226 50.2%	224 49.8%

**Table 2 diagnostics-12-00563-t002:** Average percentage changes in the intervertebral foramen area by side and the level of the spine.

Level	Right Side				Left Side			
	Mean Percentage Changes of the Intervertebral Foramen Area (%)	Min.	Max.	Std. Deviation	Mean Percentage Changes of the Intervertebral Foramen Area (%)	Min.	Max.	Std. Deviation
L1–L2	−4.65	−20.1	9.80	6.36	−4.70	−34.4	31.4	8.53
L2–L3	−6.93	−29.4	16.2	8.58	−6.93	−25.9	9.2	6.82
L3–L4	−6.27	−38.5	12.3	7.91	−4.32	−29.3	19.6	8.18
L4–L5	−3.65	−22.6	27.6	9.15	−2.60	−31.3	35.7	9.53
L5–S1	2.48	−25.4	45.4	11.48	2.25	−20.4	30.5	10.1

**Table 3 diagnostics-12-00563-t003:** Categorical variable information.

Categorical Variable		*n*	Percent of Cases
Dermatomal pain (*Dependent Variable*)	0 (No Pain)	492	54.7%
	1 (Pain)	408	45.3%
Mean percentage changes of the area of the intervertebral foramen (%) (*Factor*)	<−7	293	32.6%
	(–7, 0)	315	35.0%
	≥0	292	32.4%

**Table 4 diagnostics-12-00563-t004:** Parameters of the generalized linear model for the “dermatomal pain” dependent variable.

Parameters		B	SE	Wald Test	*p*	Odds Ratio	95%CI	
Side	Right	0				1		
	Left	0.463	0.177	6.88	0.009	1.59	1.12	2.25
Level	L1–L2	0				1		
	L2–L3	0.803	0.215	13.9	<0.001	2.23	1.46	3.40
	L3–L4	1.46	0.265	30.4	<0.001	4.30	2.56	7.22
	L4–L5	2.07	0.289	51.2	<0.001	7.89	4.48	13.9
	L5–S1	2.39	0.311	59.2	<0.001	11.0	5.96	20.2
Mean percentage changes of the intervertebral foramen area (%)	≥0	0				1		
	(–7, 0)	−0.212	0.215	0.971	0.324	0.809	0.531	1.23
	<−7	−0.103	0.203	0.257	0.612	0.902	0.605	1.34
(Intercept)		−1.71	0.326	27.5	0.000	0.181	0.096	0.343

B, regression coefficient; SE, standard error; CI, confidence interval.

## Data Availability

The dataset analyzed are not publicly available but are available from the corresponding author on reasonable request.

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
