# Peer review of "Associations between Patient Report of Pain and Intervertebral Foramina Changes Visible on Axial-Loaded Lumbar Magnetic Resonance Imaging"

_diagnostics, 2022, doi:10.3390/diagnostics12030563_

Round 1

Reviewer 1 Report

For many years, standing simulation systems have been proposed in patients undergoing spine MRI.
However, there is never proof that the simulation systems reproduce the physiological condition of the patient (orthostatism) with good approximation.

Do you think you have solved this problem? As ?

Author Response

Response to Reviewer 1 Comments

Dear Reviewer #1,

Authors are thankful for giving us the opportunity to submit a revised manuscript entitled “Associations Between Patient Report of Pain and Intervertebral Foramina Changes Visible on Axial-Loaded Lumbar Magnetic Resonance Imaging”. We appreciate the time and effort you have dedicated to providing your valuable feedback on our manuscript. We are grateful for your insightful comments on our paper. We have been able to incorporate changes to reflect most of the suggestions provided. We have highlighted the changes within the manuscript using the "Track Changes" function in Microsoft Word.

Here is a point-by-point response to your comments and concerns.

Point 1: For many years, standing simulation systems have been proposed in patients undergoing spine MRI. However, there is never proof that the simulation systems reproduce the physiological condition of the patient (orthostatism) with good approximation. Do you think you have solved this problem? As?

Response 1: High-field-strength magnetic resonance (1.5 and 3.0T) scanners offer a great deal of value to patient care. Lumbar spine MRI is one of the most frequently performed examinations of all MRIs, but MRI images, obtained during the unloaded recumbent position, do not correlate significantly with the causes of back pain. The classic works of Nachemson from the early 1960s and Rohlmann from the 2010s showed that the highest degree of intradiscal pressure in the lumbar spine occurs in standing and sitting positions. Because intradiscal pressure is lower when an individual is lying than in sitting and standing positions, we should expect that standing may lead to changes in the foraminal area. Upright magnetic resonance imaging would be a theoretically ideal diagnostic tool to simulate the spinal column under physiological conditions, but these systems are recently only low-field MRI (0.3T), which provides low image quality with low signal-to-noise ratio and errors due to motion artifacts or limited space for patients. An attempt to overcome the MRI unload caused by the patient being in a lying position is the incorporation of a high-resolution examination performed with high-field-strength MRI scanners in the supine position with axial loading, simulating physiological loading. In our opinion, this is non-ideal but the most comparable examination to the original physiological state today. Another attempt to resolve the unloaded-loaded problem is waiting for the invention of high-field upright MRI.

Reviewer 2 Report

The study was conducted on 90 patients who received MRI of the lumbar spine due to low back pain; it aimed to correlate the lumbar MRI unloaded-loaded foraminal area changes with dermatomal pain in the patient's pain drawings. Dynamic changes of the dermatomal pain distribution related to the intervertebral foramen area changes between quantitative conventional supine MRI (unloaded MRI) and axial loading MRI (alMRI) were
 analyzed.The most significant  effect of reducing the foraminal area (-6.9%) was reported at levels of L2-L3 in axial loading images. The incidence of pain in the dermatomes increases linearly with the spine level, from 15.6% at L1 to 63.3% at L5 on the right and from 18.9% at L1 to 76.7% at L5 on the left. 
Differences in area changes between foramen assigned to painful dermatomes and foramen assigned to nonpainful dermatomes were not significant. 

There are several important issues with this study - retrospective character, low number of patients and the technique of interrogation of the neuroforamina -  axial loading is probably not the clinical mechanism of the foraminal pathology,  but flexion and extension of the spine and its influence on intervertebral foramina.This aspect has been discussed in the Discussion section. Further important issue is that the neurological examination of 90 patients is missing - I suggest to include how many of these patients did have deficits and how many of these patients had the L1-L2-L3-L4-or L5 radiculopathy - I suggest that the interpretation of the changes in neuroforaminal area is correlated to the clinic. Otherwise it is unclear - how many patients had an isolated radiculopathy; how many patients had overlapping problems; how many of these 90 patients had actually a neuroradiologically relevat findings; how many underwent conservative therapy prior and how many following the MRI examination; and were there any operatively treated cases. I find this crucial, since the cases where a patient has isolated radiculopathy only due to foraminal stenosis or intraforaminal herniated disc are not so common. The figures which were presented show patients who don´t seem to have severe degenerative pathology. Please provide several illustrative examples; if these are the patients without severe degeneration or findings which prompted surgery or intensive conservative treatment, than this should be clearly stated. 

Include a figure depicting the technique of axial loading MRI - figure or photo depicting the technique. 

Furthermore, in Conclusions authors claim "This morphometric and clinical information may help minimize the incidence of injury to the lumbar nerve root during endoscopic foraminal approaches". Foraminal stenosis is one important part of the puzzle, especially in the failed back surgery, but I urge authors to explain on what grounds do they expect these axial loading MRI information ot reduce incidence of injury. Either to explain it, or to leave this sentence out. 

Author Response

Response to Reviewer 2 Comments

Dear Reviewer #2,

The authors are thankful for giving us the opportunity to submit a revised manuscript entitled “Associations Between Patient Report of Pain and Intervertebral Foramina Changes Visible on Axial-Loaded Lumbar Magnetic Resonance Imaging”. We appreciate the time and effort you have dedicated to providing your valuable feedback on our manuscript. We are grateful for your insightful comments on our paper. We have been able to incorporate changes to reflect most of the suggestions provided. We have highlighted the changes within the manuscript using the "Track Changes" function in Microsoft Word.

Here is a point-by-point response to your comments and concerns.

Point 1: The study was conducted on 90 patients who received MRI of the lumbar spine due to low back pain; it aimed to correlate the lumbar MRI unloaded-loaded foraminal area changes with dermatomal pain in the patient's pain drawings. Dynamic changes of the dermatomal pain distribution related to the intervertebral foramen area changes between quantitative conventional supine MRI (unloaded MRI) and axial loading MRI (alMRI) were analyzed.The most significant  effect of reducing the foraminal area (-6.9%) was reported at levels of L2-L3 in axial loading images. The incidence of pain in the dermatomes increases linearly with the spine level, from 15.6% at L1 to 63.3% at L5 on the right and from 18.9% at L1 to 76.7% at L5 on the left. Differences in area changes between foramen assigned to painful dermatomes and foramen assigned to nonpainful dermatomes were not significant. There are several important issues with this study - retrospective character, low number of patients and the technique of interrogation of the neuroforamina -  axial loading is probably not the clinical mechanism of the foraminal pathology,  but flexion and extension of the spine and its influence on intervertebral foramina.This aspect has been discussed in the Discussion section. Further important issue is that the neurological examination of 90 patients is missing - I suggest to include how many of these patients did have deficits and how many of these patients had the L1-L2-L3-L4-or L5 radiculopathy - I suggest that the interpretation of the changes in neuroforaminal area is correlated to the clinic. Otherwise it is unclear - how many patients had an isolated radiculopathy; how many patients had overlapping problems; how many of these 90 patients had actually a neuroradiologically relevat findings; how many underwent conservative therapy prior and how many following the MRI examination; and were there any operatively treated cases. I find this crucial, since the cases where a patient has isolated radiculopathy only due to foraminal stenosis or intraforaminal herniated disc are not so common.

Response 1: We appreciate the reviewer’s insightful analysis. We enrolled 90 consecutive patients referred to our department for a lumbar spine MRI after referral physicians performed the clinical assessment with a focus on low back pain. No strict neurological examinations were performed at the time of the MRI. We have included this in the limitations. Patients were asked to complete Pain Drawings at the time of MRI. Pain assessment is described in methods. We believe that our work may help to understand the physiology of the spine exposed to external forces and discover the relationship between the imaging examination results and the perceived symptoms. We excluded patients with previous lumbar surgery, acute trauma, spinal malignancies, or spinal infection. We have included this in the manuscript. We have not performed follow-up or longitudinal assessment, but we appreciate the Reviewer's insightful suggestion and acknowledge that it would be helpful in future research questions.

Point 2: The figures which were presented show patients who don´t seem to have severe degenerative pathology. Please provide several illustrative examples; if these are the patients without severe degeneration or findings which prompted surgery or intensive conservative treatment, than this should be clearly stated.

Response 2: Presented patients had moderate degenerative pathology (intervertebral disc degeneration according to the Pfirrmann classification - grades 3-5; facet joint degeneration according to Weishaupt classification – grades 0-2; lumbar spinal canal stenosis according to the Schizas classification – grades A2-B; disc herniation according to the Michigan State University classification – grades 0-2a; foraminal stenosis according to Lee classification – grades 0-2). We noticed that patients with the most severe degenerative pathology appeared to have minor changes in the foraminal area (an additional study must confirm these observations), making the figures with severe degenerative pathology less understandable for the readers. Nevertheless, we presented figures with the most representative and apparent changes to familiarize readers with this MRI technique. The study design did not include follow-up or longitudinal observation, so the study could not answer the consequent surgery or intensive conservative treatment. Still, we appreciate the Reviewer's insightful suggestion and acknowledge that it would be helpful for future research questions.  Our study aimed to investigate the relationship between dermatomal pain and changes in the foraminal area, which would be the first published report in the literature on this topic.

Point 3: Include a figure depicting the technique of axial loading MRI - figure or photo depicting the technique.

Response 3: We appreciate the Reviewer's insightful suggestion. We have included a figure depicting the technique of axial loading MRI.

Point 4: Furthermore, in Conclusions authors claim "This morphometric and clinical information may help minimize the incidence of injury to the lumbar nerve root during endoscopic foraminal approaches". Foraminal stenosis is one important part of the puzzle, especially in the failed back surgery, but I urge authors to explain on what grounds do they expect these axial loading MRI information ot reduce incidence of injury. Either to explain it, or to leave this sentence out.

Response 4: As suggested by the Reviewer, we have removed this sentence from the conclusions of our manuscript.

Round 2

Reviewer 1 Report

Nothing

Reviewer 2 Report

The authors have sufficiently explained the reviewers remarks.